# Evaluation of Fatigue Life of Recycled Opaque PET from Household Milk Bottle Wastes

**DOI:** 10.3390/polym14173466

**Published:** 2022-08-25

**Authors:** Adrian Korycki, Christian Garnier, Silvia Irusta, France Chabert

**Affiliations:** 1Laboratoire Génie de Production, Ecole Nationale d’Ingénieurs de Tarbes, Institut Polytechnique de Toulouse, Université de Toulouse, 47 Avenue d’Azereix, 65016 Tarbes, France; 2Department of Chemical and Environmental Engineering, Institute of Nanoscience of Aragon, University of Zaragoza, Mariano Esquillor s/n, 50018 Zaragoza, Spain

**Keywords:** recycling, opaque PET, mechanical properties, fatigue, fracture

## Abstract

Polyethylene terephthalate (PET) is among the most used thermoplastic polymers in large scale manufacturing. Opaque PET is increasingly used in milk bottles to save weight and to bring a glossy white aspect due to TiO_2_ nanoparticles. The recyclability of opaque PET is an issue: whereas the recycling channels are well established for transparent PET, the presence of opaque PET in household wastes weakens those channels: opaque bottles cannot be mixed with transparent ones because the resulting blend is not transparent anymore. Many research efforts focus on the possibility to turn opaque PET into resources, as one key to a more circular economy. A recent study has demonstrated the improvement of the mechanical properties of recycled PET through reactive extrusion. In the present work, the lifespan of recycled opaque PET has been evaluated throughout tensile–tensile fatigue loading cycles at various steps of the recycling process: The specimens are obtained from flakes after grinding PET wastes (F-r-OPET), from a subsequent homogenization step (r-OPET-hom) and after reactive extrusion (Rex-r-OPET). Virgin PET is also considered as a comparison. First, tensile tests monitored by digital image correlation have been carried out to obtain the elastic modulus and ultimate tensile stress of each type of PET. The fatigue properties of reactive REx-r-OPET increase, probably associated with the rise of cross-linking and branching rates. The fatigue lifespan increases with the macromolecular weight. The fracture surface analysis of specimens brings new insight regarding the factors governing the fatigue behavior and the damaging mode of recycled PET. TiO_2_ nanoparticles act as stress concentrators, contributing to void formation at multiple sites and thus promoting the fracture process. Finally, the fatigue life of REx-r-OPET is comparable to those of virgin PET. Upcycling opaque PET by reactive extrusion may be a relevant new route to absorb some of the growing amounts of PET worldwide.

## 1. Introduction

According to the latest report by Plastics Europe [1], in 2020, almost 55 million tons of plastics were produced in Europe, which is 15% of world production. At the same time, the global production of polymers reached 367 million tons and increased by almost 2.5% each year until 2019. The market is mainly stimulated by the demand generated by the packaging industry (40.5%), building and construction (20.4%), and automotive sectors (8.8%). In 2020, more than 29 million tonnes of plastic post-consumer waste were collected in Europe. More than one-third of the collected plastic was sent to recycling facilities, but over 23% was still sent to landfills and more than 40% to energy recovery operations, as seen in Figure 1. The COVID-19 pandemic contributed to an increase the plastic production through the fabrication of masks, disposable workwear, and packaging for various goods.

Researchers have estimated that up to 12 million tons of plastic end up in the ocean every year, which is the equivalent of a rubbish truck every single minute [2]. In landfills, they accumulate rather than decompose. Recycling plastics is one of the many initiatives launched in Europe to turn waste into resources to create a circular economy for plastics. In the last decades, increasing interest has been focused on recycling plastic wastes [3,4,5,6].

Polyethylene terephthalate is one of the big six thermoplastics dominating the market. It is an acknowledged polyester with high mechanical strength, chemical inertness, and high dimensional stability [7]. The popularity of PET as a packaging material comes from several properties, especially its glass-like transparency and low weight, which, combined with flexibility and mechanical resistance, make packaging break-resistant [8,9]. It is a good barrier to gases, which enables the use of PET in the production of packaging intended for packing and storing carbonated beverages. PET can also be used in combination with other materials, including textile raw materials, which increases the strength of fabrics [10,11]. According to the Eunomia’s report in 2022 [12], the consumption of PET in Europe is dominated by beverage bottles (47% of overall EU PET demand), other single-use packaging (20%), and textiles and fibres (33%). Figure 2 represents market shares of PET in Europe. Focusing on single use bottles, 78% of them are made with clear PET, 20% from coloured transparent PET, and only 2% are opaque PET. Despite this low amount of 2%, the presence of opaque PET disturbs the existing recycling channels of PET as developed below.

PET is not biodegradable, contributing to pollution when it is thrown in nature. However, PET can be recycled, like almost all thermoplastics. Plastic recycling refers to operations that aim to recover plastic that can be converted into new items as a substituted version of virgin plastic. Recycling PET can help achieve a balance in the ecosystem severely disturbed by excess plastic production. Whereas transparent PET (78% of market shares) or coloured PET (20% of the market) can be fully recycled, the release of opaque PET bottles (2% of the market) a few years ago disturbed the PET recycling channels. Indeed, recycling companies are not well-prepared to separate and recycle it, unlike with transparent PET. Indeed, the recycling channels for transparent PET are well established. Opaque PET cannot be tolerated in the loop of transparent, colourless PET recycling since the latter is recycled almost uniquely into transparent sheets and bottles. Opaque PET may be incorporated into coloured PET up to 20–30% without noticeable loss of colour or processability [13,14], but this quantity is not enough to absorb the increasing amount of opaque PET launched on the market. Reduction of the molecular weight during extrusion and injection moulding is one of the main issues in PET recycling [15]. Indeed, the sensitivity of its chemical backbone to hydrolysis induces chain breakage when virgin or recycled PET are processed with improper drying.

PET is so sensitive to hydrolysis that an alternative method of recycling has been developed to recover initial terephthalic acid (TPA) monomers. It can be performed in mild conditions and it can tolerate highly contaminated post-consumer wastes [16]. Acidic and neutral hydrolysis gives high yields of TPA. All mechanical impurities present in the polymer stay in the TPA, as such affecting the purity of TPA. Alkaline hydrolysis seems to be the best option among chemical recycling PET methods [17]. The so-obtained monomers can be turned into virgin PET by polymerization. Some attempts to industrialize such methods are being made however, the investment cost and the energy consumption are high. For these reasons, mechanical recycling through extrusion is still worth to be developed. Many applications are foreseen from recycled opaque PET in structural applications, insulating foams and furniture. For that, recycled PET is expected to reach similar mechanical properties to virgin PET. Whereas the mechanical properties are usually studied through static characterization such as tensile and flexural tests, fatigue tests are crucial to assess the reliability of the recycled material whatever the targeted application.

Material fatigue is measured through cyclic load at a fixed frequency. Fatigue fracture generally occurs through a two-step process, initiation and propagation stage. As a consequence of high values of fatigue loads, some microstructural and morphological changes can have manifested into the polymers. The first step involves the initiation of microcracks or other damages at inhomogeneities or defects in the material. This damage can initiate and evolve at nominal stress levels far below the yield or tensile strength of the material. The second step involves the growth of damage through the coalescence of microcracks and the subcritical propagation of these small cracks to form large cracks and ultimately cause material failure. Fatigue studies are numerous for metals, while they are scarce for plastics, composites, and ceramics [18].

For most polymeric materials, the initiation time can be orders of magnitude greater than the propagation time [19]. Changes in mechanical behaviour during fatigue of thermoplastic, such as polyamide, polyacetal, polypropylene, polycarbonate, poly(methyl methacrylate), polyethylene, or polyethylene terephthalate, have been reported in the literature [20,21,22,23].

Fatigue tests are generally performed in compression, tension/compression, or tension configurations. Fatigue tests consist of applying cyclic loading and unloading up to specimen crack to measure the number of cycles that the material can withstand. These tests are used either to generate fatigue life and crack growth data, identify critical locations, or demonstrate the safety of a structure that may be susceptible to fatigue [19]. Tensile-tensile configuration is the most used for studying the fatigue behaviour of thermoplastics. The tensile-tensile fatigue behaviour of two semicrystalline thermoplastics: polyacetal (Celcon^®^ M90 from Celanese ™, Dallas, Texas, USA) and polyamide 6.6 (Zytel^®^ 101, DuPont™, Wilmington, DE, USA) were studied by Lesser [20]. The correlation with dynamic viscoelastic response shows that stress softening and hardening can occur in load cyclic conditions. Both materials display a thermally dominated region at higher stress levels, approximately 50 MPa for the polyacetal and 40 MPa for the polyamide 6.6. A mechanically dominated region resides at lower levels where the polyamide demonstrates better fatigue resistance when compared to polyacetal. At a stress level of 30 MPa, the polyamide can sustain more than 2 × 10^7^ cycles before failure and polyacetal will last 5 × 10^6^ cycles. During the initiation stage, both materials become less viscoelastic and more elastic while the propagation stage contribution is minor. The crack initiation time has been reported as the majority of the fatigue life. A wide-ranging work on fatigue behaviour on polypropylene, including the influence of temperature and mean stress effects, was presented by Mellott and Fatemi [22]. The fatigue tests were performed at room temperature, −40 °C, and 85 °C. The effect of mean stress was modelled using various parameters, the most effective model turned out to be the Walker mean stress model. In this model, the mean stress effect depends on the material mean stress sensitivity factor and vary with temperature. It predicts this effect very well for the mean stress sensitivity factor at 0.5. In both low and high cycle fatigue, the stress levels used to produce a similar lifespan to the room temperature fatigue test were about 3 times lower at 85 °C and about 2.5 times higher at −40 °C. This shows that the fatigue behaviour of the polymer is very sensitive to temperature.

In a study by Janssen et al. [21] of fatigue behaviour, several thermoplastics (PC, PMMA, HDPE, and iPP) specimens were manufactured. According to the analytical method, which is able to perform quantitative fatigue life predictions from a set of creep life data, the fatigue life depends on the stress amplitude of the cyclic signal and the stress dependency of material. Amplitude form indicated as a square wave has a higher acceleration of plastic flow than a sinusoidal or triangular wave. Thus, the lifetime is lower under a square wave than under a triangular wave. The fatigue failure and fracture morphologies of fibres, recycled and virgin PET blends were studied by Elamri et al. [23]. Their results confirmed that recycled/virgin PET fibres could be used in the same applications as those from virgin PET polymers (5.1 × 10^5^ for 10/90 of recycled/virgin PET) and would achieve a good resistance level to cyclic loading solicitation (4.1 × 10^5^ for 75/25 of recycled/virgin PET) at 70% of ultimate tensile stress (UTS) of the mean tensile breaking stress. The mean lifetime for recycled PET was registered at 2.8 × 10^5^ cycles. The failure occurs more frequently at the maximum load of 75%UTS. Thus, the damages developed by the interaction of thermal and fatigue loadings can be different from the damage for each type of loading separately. In most cases, fatigue strength is used as the maximum value of stress that a material can withstand for a specified number of cycles without failure. The fatigue limit has been shown to reflect the intrinsic strength of the covalent bonds in polymer chains that must be ruptured in order to extend a crack.

The fatigue life and its limits strongly depend on process condition, chemical structure, morphology, crystallization, density and porosity, as well as the presence of nanoparticles [19,24,25,26]. It was shown by Sauer and Richardson [20] on polystyrene that as molecular weight increases, the S-N (stress-fatigue life) curves shift to higher stress values and a higher number of cycles, increasing the fatigue limit. Kim et al. [25] have studied the influence of density and porosity on polyamide 12. The results show that fatigue life decreases as the diameter of the pores became larger, and increases as their aspect ratio increases, which is the ratio of the maximal and minimal diameter of the pores. The presence of nanoparticles, such as CaCO_3_ in high-density polyethylene, has been observed by Khan [26]. It was noted that particles act as stress concentrators, helping in the void formation, which indicates that crack growth occurs through a fully developed crazing process.

A small number of studies have been devoted to the effect of fatigue behaviour of recycled opaque PET. A previous work of Tramis et al. [10] focused on the fatigue life of recycled polypropylene/recycled opaque PET blends. He showed that the fatigue life of PP/PET blends increases with PET content for the lowest %UTS. Thus, PET demonstrates the longest fatigue life, and damaging occurs sooner for low PET content.

Candal et al. [27] have recently demonstrated that recycled opaque PET can be regenerated by chain extension. After reactive extrusion, the material is named REx-r-OPET. Based on the characterization of REx-r-OPET, she concluded that the reactive extrusion process is a way to upcycle opaque PET into a material with static mechanical properties that are comparable to those of a typical virgin PET. In this work, the mechanical properties of recycled opaque PET collected from household wastes have been studied. The tensile and fatigue properties of so-called reactive extruded REx-r-OPET are studied and compared to recycled opaque PET (without reactive extrusion step) and virgin PET. Our work aims to assess their fatigue life and to correlate the lifespan to the fracture mechanisms by analyzing the morphology of fractured surfaces.

## 2. Materials and Methods

### 2.1. Materials

Four different PET materials were studied. Recycled opaque PET, referred as r-OPET, was supplied by Suez RV Plastiques Atlantique, Bayonne, France, under the tradename Floreal in flakes. They were obtained by grinding post-consumer opaque UHT milk bottles. The types of PET and the sample names are presented in Figure 3. All the tested samples were injection moulded by IPREM, UPPA, Pau, France.

F-r-OPET are the specimens injected directly from flakes;r-OPET-hom are made by processing the flakes via extrusion in filament followed by a pelletizing step. An extruder with L/D = 25 (IQAP-LAP E30/25, Masies de Roda, Spain) was employed with four heating zones along the profile of the screw. The temperature profile was set to 175 (hopper zone)/195/225/245 °C (die zone) and a screw rotation speed of 50 rpm. The process was performed in a N_2_-controlled atmosphere to minimize thermo-oxidative degradation. The r-OPET-hom filament obtained was quenched in room temperature water bath, dried, and then cut into pellets. Then, this material was recrystallized in an oven at 120 °C for 4 h to increase the crystallinity up to 20–30 %. This step results in homogenizing the recycled material. Then, the pellets were injection moulded [27].REx-r-OPET are obtained by processing the previous pellets by reactive extrusion with the addition of a styrene-acrylic multi-functional epoxide oligomer commercially available by BASF (Ludwigshafen, Germany) under the brand name Joncryl ADR-4400. The reactive extrusion of REx-r-OPET was performed using a corotating twin-screw extruder with L/D = 36 (KNETER-25X24D, Collin GmbH, Maitenbeth, Germany). As a reactive (chain extender) reagent, a multifunctional epoxide agent with an epoxy equivalent weight of 485 g.mol^−1^ and functionality of 14 was added (1 wt%). The temperature profile of the extruder was set to 175 (hopper zone)/215/230/235/240/245/245 °C (die zone), and the screw speed was 40 rpm, leading to residence times of 4.1 min. The process was performed in vacuum to avoid further degradation. Then, the REx-r-OPET product was water-cooled, dried, and pelletized, after which the acquired material was once again recrystallized at 120 °C for 4 h. This step was done by Centre Català del Plàstic—Universitat Politècnica de Catalunya Barcelona Tech (EEBE-UPC)-ePLASCOM Research Group, Barcelona, Spain. The process of reactive extrusion is described in Candal et al. [27];v-PET is a virgin PET supplied by Novapet (Zaragoza, Spain). Contrary to recycled opaque PET, virgin PET does not contain TiO_2_ particles.

### 2.2. Methods

#### 2.2.1. Tensile Test

Tensile tests were performed following ISO 527-1 and ISO 527-2 standards, in a thermoregulated room at 23 °C, using an electromechanical tensile test device Instron 5500R, Norwood, Massachusetts, USA, equipped with a 5 kN load cell. A displacement speed of 5 mm min^−1^ was applied. Load and displacement of the crossbar are recorded during the tests. The ultimate tensile strength (UTS) is defined as the maximum load a sample reached during a tensile test. The elastic modulus was computed from the stress-strain curve as the slope in the strain intervals 0.0005 and 0.0025, as recommended in the standard ISO 527-1.

#### 2.2.2. Fatigue Test

In mechanically induced uniaxial fatigue tests, the stress or strain has oscillated about the same mean stress (strain) value. The tests are performed following the ISO 13003:2003 standard, in a load-control mode on a servo-hydraulic fatigue machine, Schenck, Darmstadt, Germany, retrofitted Instron, Norwood, Massachusetts, USA, equipped with a 32 kN load cell. All the specimens are loaded by a 5 Hz sinusoidal waveform at a constant loading amplitude based on tensile test data. A PID loop is used to ensure that the applied load respect the theoretical set point. The parameters based on proportional, integral, derivative terms (P, I and D respectively) and delay are first defined with the tested material. Then, the load ratio (R) is defined as the ratio of the minimum to the maximum tensile stress (σ), as follows:(1)$$R=\frac{\sigma_{\min}}{\sigma_{\max}}\;$$

With σ_max_ and σ_min_ being defined respectively as follows:(2)$$\sigma_{\max}=\frac{\%\mathrm{UTS}*\mathrm{UTS}}{100},\;\sigma_{\min}={R*\sigma }_{\max}$$

The ratio R was set at 0.1 for all experiments up to the standard ISO13003:2003. For each set of materials, the levels of fatigue load were applied, from 95%UTS to 70%UTS, by step of 5%. Tension-tension load control fatigue tests were carried out.

#### 2.2.3. Digital Image Correlation

Aramis 2 M cameras benchmark coupled to the GOM^©^ Aramis digital image correlation (DIC) software, Braunschweig, Germany, is used to record displacements of the samples. A 50 mm focal length sensor is used. A reference image is recorded just before launching the tensile tests. Images are acquired at a rate of 5 Hz with a facet field of 15 × 10 pixels, and a spatial resolution of 60 μm, in both directions. Displacements are deduced by subtracting the reference image from the subsequent images. Computation led to the deformation samples withstood during tensile tests.

#### 2.2.4. Backscattered Scanning Electron Microscopy

The morphology of cryo-fractured and fatigue fractured surfaces of samples are analysed by Inspect F50 (FEI, Hillsboro, OR, USA) field emission gun scanning electron microscope (SEM) operated at 5–10 kV. Samples are previously carbon-coated (Leica Microsystems GmbH, Wetzlar, Germany EM ACE200 coater) to allow backscattered electrons observations.

## 3. Results and Discussion

### 3.1. Tensile Properties

Tensile test runs and post-treatment of DIC images were done for at least 3 samples. Two methods were used for the calculation: stress–strain curves from tensile properties (named Meca) and more accurate digital image correlation (named DIC). A slight difference between both results is noticed in Figure 4. The stress–strain curves of the PET specimens show a non-linear behaviour with a first linear step followed by a strain-softening region. The strain at break has not been indicated, and all specimens exhibit extensive ductility. Comparing the stress–strain curves of all materials, the results are quite reproducible.

The ultimate tensile strength was extracted from the strain–stress curves as the maximum stress recorded during the tensile test. Each bar indicates the mean value and the dots are for each test. As expected, the highest UTS was obtained for virgin PET at 51.5 MPa. Then, REx-r-OPET has a higher UTS than r-PET-hom (Figure 5, on the left). However, by considering the standard deviation, the results of all recycled opaque PET are comparable. The difference between REx-r-OPET and virgin PET is only 1.7 MPa.

The tensile modulus has been calculated for a deformation between 0.0005 and 0.0025, based on DIC measurements. Figure 5, on the right, shows the results and exhibits the REx-r-OPET has a slightly higher tensile modulus than other PET materials. It can also be noticed that v-PET, r-OPET-hom and F-r-OPET have a very similar tensile modulus (around 2175 MPa). It is worth noticing that for F-r-OPET, the standard deviation is higher than for the others. F-r-OPET material is obtained by extrusion of r-OPET flakes without the homogenization step, so, we expected such variation from one specimen to another.

As shown in Table 1, the summary of the tensile results is very similar for all the PET materials. The UTS is slightly lower for the recycled opaque PET than for v-PET. However, the elastic modulus has been improved for REx-r-OPET due to an increase in chain length caused by the action of the chain extender during reactive extrusion. For the next step, the relevant data for fatigue tests are UTS.

### 3.2. Fatigue Properties

From fatigue tests, the number of loading cycles to cause failure was recorded for a given %UTS, which allowed to draw the S-N curve. It is plotted with the cyclic stress (S) against the cycles to failure (N) on a semi-logarithmic scale. Fatigue life and fatigue limit may be highlighted from those curves. The amplitude (A) of the fatigue load is then defined by the difference between the maximum (or minimum) and the average load. The type of cyclic is typically a sinusoidal wave with the amplitude, mean stress, and frequency all adjustable parameters to consider, as shown in Figure 6.

The transition from thermal to mechanical failure has been described in terms of a changeover stress level, which depends on test frequency, mean stress, cyclic waveform, and specimen surface-area-to-volume ratio. Adjustment of the experimental data by a logarithmic law (Equation (3)) represents fatigue life and may point towards different damaging mechanisms occurring at low or high cycle counts for the materials.
(3)$$S=\mathrm{alog}\left(N\right)+b\;$$
where a and b represent kinetics of damage apparition. They may give insights on change when comparing different materials. When the value of a is lower, a faster loss of fatigue properties is observed. It can be said that fatigue life is shorter.

In the case of a large number of cycles during a fatigue test, the critical stress (*σ_α_*) versus the number of load reversals for failure (*N_r_*) is fully reversed. Constant amplitude fatigue can be described by the Basquin model [28] presented in Equation (4).
(4)$$\sigma_{\alpha }=\frac{\Delta \sigma }{2}=\sigma_{f}N_{r}^{b},\;\;N_{r}=2\left(N_{f}-1\right)\approx 2N_{f}\;$$
where *σ_f_* is the fatigue strength coefficient and *b* is the fatigue strength exponent. Those two values are the intercept and slope of the linear least-squares fit to stress amplitude, Δ*σ*/2, versus reversals to failure, 2*N_f_*, using a semi-logarithmic scale. The number of reversals to failure is related to the number of cycles.

The fatigue limit is the stress level below which an infinite number of loading cycles can be applied to a material without causing fatigue failure. In this study, the number of cycles to failure has been set up as one million.

The stress-failure curves (S-N curves) are derived from tests on samples where regular sinusoidal stress was applied by a testing machine and counts the number of cycles to failure. The results are reported in Figure 7. On the right, each coloured point assesses a specimen. The dispersion of results is high: this is common in fatigue tests. A sample is considered “fatigue resistant” if it can withstand a million cycles without breaking. Generally, the damage mechanism occurs at high cycle counts. For some %UTS, some specimens did not break, and the experiments were stopped. They are indicated with black arrows in Figure 7, on the left. On the right, a black mark figures the average of at least 3 values with its standard deviation for each type of PET. As expected, the standard deviation is the highest one for PET obtained from flakes (F-r-OPET). Virgin PET withstands the highest number of cycles. For a relevant analysis of the fatigue results, we applied two existing models.

The S-N curves are drawn in Figure 8 for logarithmic law on the left and Basquin law on the right. Regarding the logarithmic law, the shape of the curve follows the usual S-N curves: as the %UTS decreases, the number of cycles at break increases. It can be seen that for 95 %UTS, the fatigue resistance of reactive extrusion recycled PET (REx-r-OPET) and virgin PET is equivalent. They are higher than those of homogenized and flakes r-OPET. The curves of the Basquin law agree with those of logarithmic law, confirming the previous conclusions.

Both presented models fit well the studied materials and are comparable. The fitting equations describing the fatigue life modelling are presented in Table 2.

Assuming that both models give similar results, where *a* in logarithmic law corresponds to *b* in Basquin law, the first law will be further investigated to extract information on the macromolecular structure of each type of material. Similar conclusions would be raised from Basquin law. Then, by considering the results of logarithmic law, the fitting parameter *a* has been gathered for all the materials and presented in Figure 9, on the left, while fatigue limits are presented on the right.

One straight line may be drawn for each material. Coefficient of determination (R^2^) is 0.95, 0.98, 0.95, and 0.94 for REx-r-OPET, r-OPET-hom, F-r-OPET, and v-PET, respectively. The fitting parameters can be split into two groups: are nearly identical for REx-r-OPET and r-OPET-hom, as well as, for a pair of F-r-OPET and v-PET.

The loss of fatigue properties is evaluated through a parameter which is the slope of the curve. This parameter shows that the fatigue properties decrease faster for REx-r-OPET with *a* = −8.1 and the homogenized one at −7.1 than for F-r-OPET and v-PET with *a* = 3.8 and 4.2 respectively. This demonstrates that the damages appear slowly in F-r-OPET and v-PET compared to REx-r-OPET and r-OPET-hom.

Besides, the fatigue limit presented in Figure 9 is 70 %UTS for r-OPET-hom and more than 85 %UTS for virgin PET. The larger the value, the more the material will be able to withstand high loads without breaking up to 1 million cycles. Despite a lack of explanation in the literature to date, we correlate this result to the macromolecular weight. The fatigue limit is shifted towards higher values with the increase of macromolecular weight. Homogenized PET, REx and virgin PET have molecular weights of 27.1 kDa, 37.3 kDa and 45.0 kDa, respectively [27]. Even if the macromolecular mass of flakes has not been measured (for lack of reproducibility from one flake to another), it is expected to be higher than those of r-OPET-hom which withstand another thermomechanical cycle during extrusion. Indeed, the recycling of PET is causing hydrolytic and thermal degradation that is responsible for its reduction in molecular weight. The presence of water promotes chain scission during extrusion processing, resulting in shorter chains with acid and hydroxyl-ester end groups. In the case of REx-r-OPET, the chain extender reacts with homogenized r-OPET through its epoxy groups. Both chain extension reactions and the generation of long-chain branching are possible [29,30,31]. As a result, an increase in the molecular weight is measured after reactive extrusion of r-OPET-hom that leads to REx-r-OPET material. The following order of macromolecular weight is proposed: r-OPET-hom < F-r-OPET < REx-r-OPET < v-PET. These results follow the same order as the fatigue limit in Figure 9, on the left. Longer macromolecular chains mean a higher entanglement rate, the latter is known to be responsible for the mechanical strength. A correlation between the a parameter and macromolecular weight and distribution has been revealed recently by Cerpentier et al. [32] on polyethylene.

Thus, we assume that throughout the fatigue test, the chains gradually disentangle and slide over each other and eventually until the chemical bonds break, which results in a decrease in fatigue properties. Moreover, the variations may be caused by porosity arising during the processing of specimens by injection moulding.

Thus, as the molecular weight increases, the fatigue limit is increased, which also results in an improvement of the fatigue resistance. REx-r-OPET for 95 %UTS can reach a similar number of cycles during fatigue test as virgin PET which is less sensitive to fatigue solicitation.

### 3.3. Fractography in Failure Analysis

Scanning electron microscopy (SEM) images of the cryogenic fracture surface of the materials are given in Figure 10 for references, and in Figure 11, the tensile–tensile stressed samples during 1 million cycles, corresponding to the last fatigue load before failure. The magnifications are ×20,000 and ×5000, respectively.

SEM images show brittle fracture with a combination of patchy type structure and fatigue striations. The presence of TiO_2_ nanoparticles is observed as spherical bright spots in recycled samples. Moreover, low particle/matrix interfacial adhesion is assumed from the spherical cavity around each mineral particle. TiO_2_ particles are not presented in virgin PET. After tensile-tensile stress is applied in Figure 10, the presence of microcavities appears. Between references and low-stress fatigued samples, no big morphological changes are seen. However, fatigue striations are visible for reactive extruded and homogenized r-OPET.

On the contrary, for a high fatigue load rate at 95 %UTS, differences are highlighted in fracture surfaces displayed in Figure 12. The fracture took place through material fibrillation for REx-r-OPET and r-OPET-hom as seen by the tortuous and fluffy surface morphology. The fibrillation of the matrix is due to thermo-mechanical solicitation during cyclic loadings. Oppositely, the fracture surface is smoother for F-r-OPET and v-PET. These observations do not correlate with the macromolecular weight, showing that another factor is predominant in the fracture behaviour. As seen in Figure 12, F-r-OPET contains numerous microvoids whereas no cavity was visible before fatigue in Figure 10. These cavities could stem from microvoids caused by injection moulding during specimen fabrication. Upon fatigue, the growth and coalescence of microvoids lead to ductile fracture.

Moreover, by looking into the SEM images in Figure 13, the presence of TiO_2_ particles is observed inside huge cavities, except for virgin PET. During ductile damage behaviour, microcracks appear in the matrix and at the interface between matrix and TiO_2_ inclusions. Microcracks nucleate around inclusions, at the matrix/TiO_2_ interface due to poor adhesion. With the rise of stresses, microcracks grow around nanoparticles. Nucleation and growth of microcracks is a thermodynamic phenomenon that produces heat. This heat is transferred to the matrix, changing its behaviour and allowing more deformation. Nanoparticles can act as stress concentrators that form the void at multiple sites and finally promote the fracture process. Elongation seems stronger around the gaps, which would support the role of particles as stress concentrators.

On the other hand, virgin PET seems to be less fibrillated. It may be supposed that since at this fatigue load, the fatigue life was longer, the macromolecules had more time to flow and to adapt to the direction of solicitation. The fact that r-OPET is found more fibrillated may be explained by higher shear flow due to thermal solicitation. When the materials are compared, again, the two main factors that are expected to affect ductility are the presence of nanoparticles in the matrix and the decrease in molecular weight. Indeed, a lower ductility of the r-OPET is expected as a consequence of the presence of nanoparticles, which is attributed to restrictions in the mobility of the matrix chains caused by the nanoparticles that promote a fracture. Moreover, lower molecular weights lead to bigger fibrillation.

## 4. Conclusions

In this work, we compared post-consumer recycled opaque PET at various steps of the recycling process. The specimens were prepared by injection moulding of grinded flakes (F-r-OPET), others were homogenized by extrusion (r-OPET-hom) and, also, modified by reactively extruding the material with a chain extender (REx-r-OPET). The reactive extrusion changed the molecular structure of the originally linear r-OPET-hom by introducing long-chain branches in the material and increasing the average molecular weight of the material. Finally, virgin PET (v-PET) were used for comparison purposes.

All tested materials show similar tensile behaviour with viscoplastic behaviour. Digital image correlation was used to obtain accurate deformation and to assess small changes in mechanical behaviour. As expected, the ultimate tensile strength of virgin PET is higher than those of recycled ones. The highest elastic modulus is for REx-r-OPET, this is explained by the chain crosslinking and branching induced by the addition of the chain extender in the reactive extrusion step.

Failure of all PET materials under cyclic fatigue follows the logarithmic law and Basquin’s equation very well. The fatigue limit can be correlated to ultimate tensile strength. Virgin PET and flakes F-r-OPET are less sensitive to fatigue solicitation and their fatigue limit are 85 %UTS and 75 %UTS respectively. An improvement of the fatigue life is obtained for reactive extruded specimens compared to other recycled ones, with a longer lifetime for the same fatigue load rates and higher fatigue limits. Moreover, it was confirmed that the factors playing a role in changes in fatigue behaviour are the presence of nanoparticles and molecular weight. Particles act as stress concentrators, contributing to void formation at multiple sites and thus promoting the fracture process. A decrease in molecular weight causes shorter fatigue life and limits.

This work demonstrates that upcyling PET from milk bottles is possible through reactive extrusion. This reactive extrusion step leads to enhanced fatigue properties compared to untreated recycled PET. Even though the properties of extruded reactive PET are still inferior to those of virgin PET, it remains sufficient to manufacture new products to help save resources and reduce the environmental impact.

## Figures and Tables

**Figure 1 polymers-14-03466-f001:**
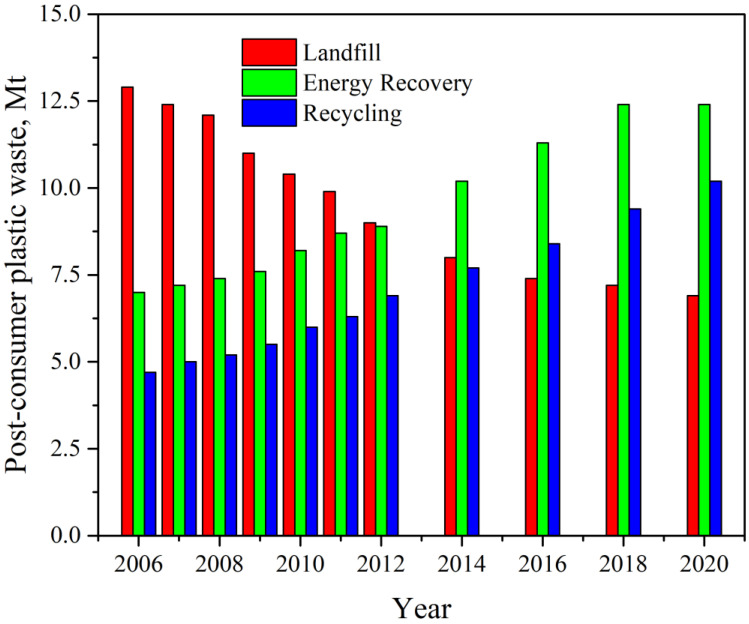
Evolution of post-consumer plastic waste treatment in Europe (red—landfill, green—energy recovery, blue—recycling) Data from [1].

**Figure 2 polymers-14-03466-f002:**
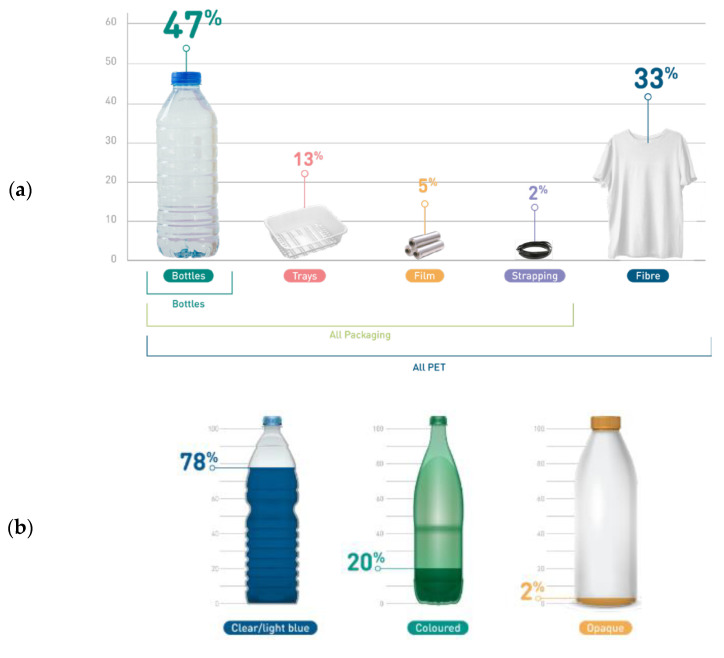
Market shares of PET (**a**) by manufacturing scope (**b**) by bottle types and colours—Reprinted with permission from Ref. [12].

**Figure 3 polymers-14-03466-f003:**
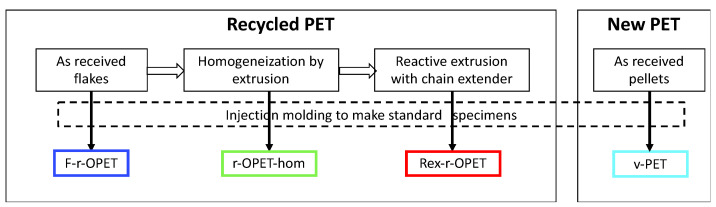
Types of PET materials used in this study.

**Figure 4 polymers-14-03466-f004:**
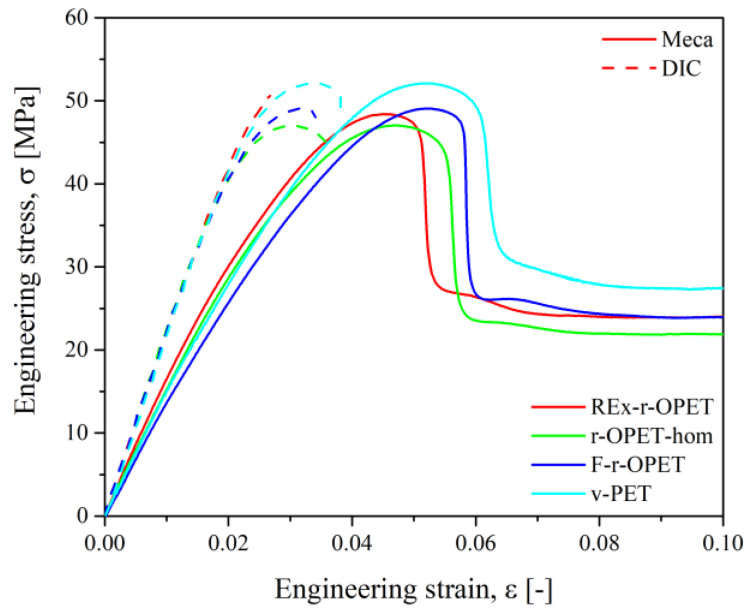
Comparison of the strain-stress curves of PET specimens, comparing the deformation obtained from tensile tests (Meca) and image treatment (DIC).

**Figure 5 polymers-14-03466-f005:**
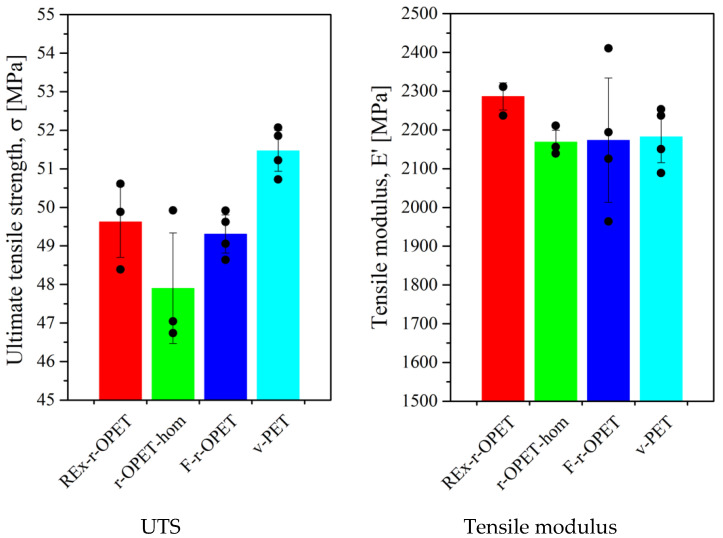
Ultimate Tensile Strength and tensile modulus properties of PET specimens, extracted from Figure 4 using DIC. Each black point corresponds to one specimen. Bars and standard deviations refer to an average of at least 3 specimens.

**Figure 6 polymers-14-03466-f006:**
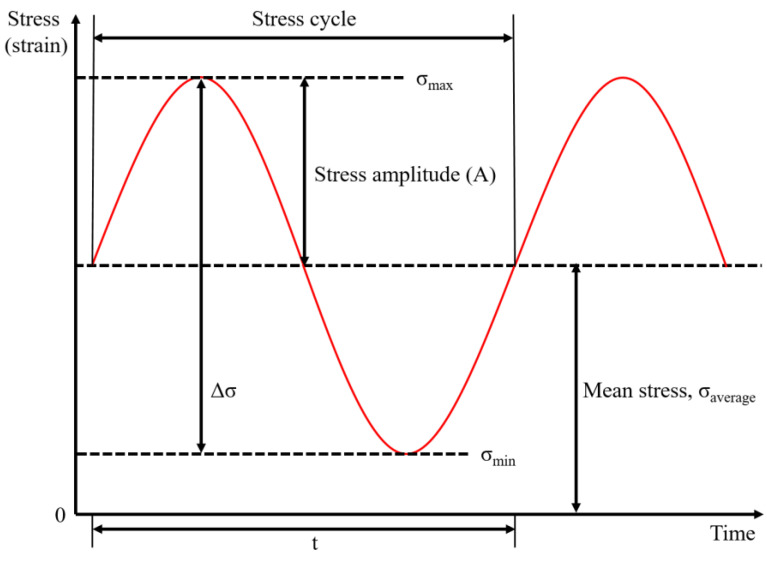
Scheme of the applied sinusoidal form during fatigue test.

**Figure 7 polymers-14-03466-f007:**
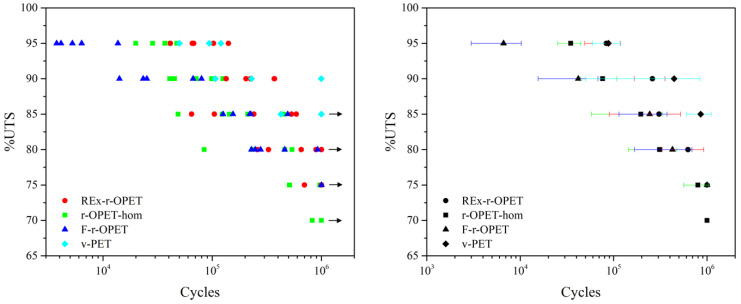
Fatigue life for various fatigue loads of PET specimens, **left**: All tested specimens (each coloured point assesses a specimen) black arrows indicate run-outs, **right**: Average of at least 3 values with its standard deviation.

**Figure 8 polymers-14-03466-f008:**
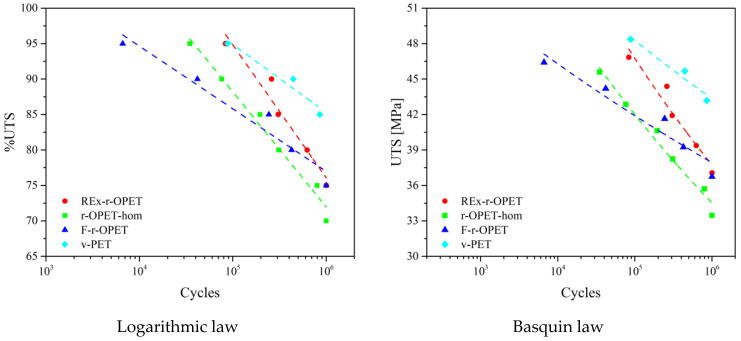
S-N curves of PET specimens fitted with logarithmic law (**left**) and Basquin law (**right**).

**Figure 9 polymers-14-03466-f009:**
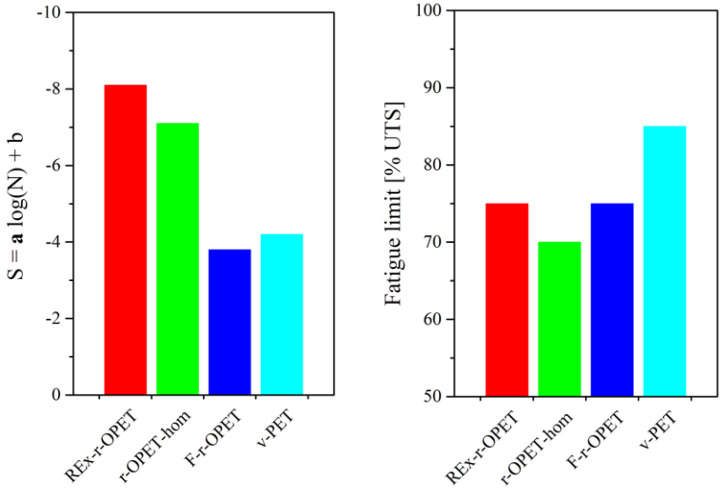
Fatigue life parameter a represents the kinetics of damage apparition (**left**) and fatigue limit (**right**) for PET specimens.

**Figure 10 polymers-14-03466-f010:**
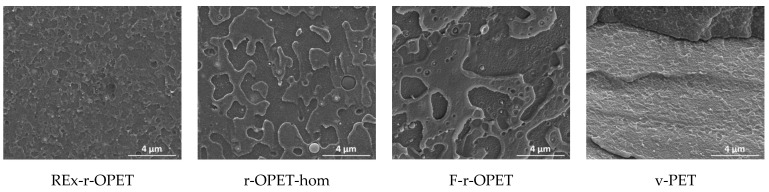
SEM images of cryogenic fracture surfaces of reference PET specimens (no fatigue).

**Figure 11 polymers-14-03466-f011:**
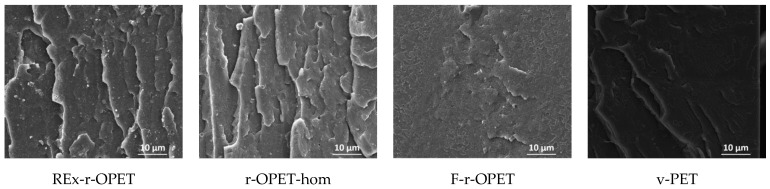
SEM images of cryogenic fracture surfaces of stressed samples during a million cycles.

**Figure 12 polymers-14-03466-f012:**
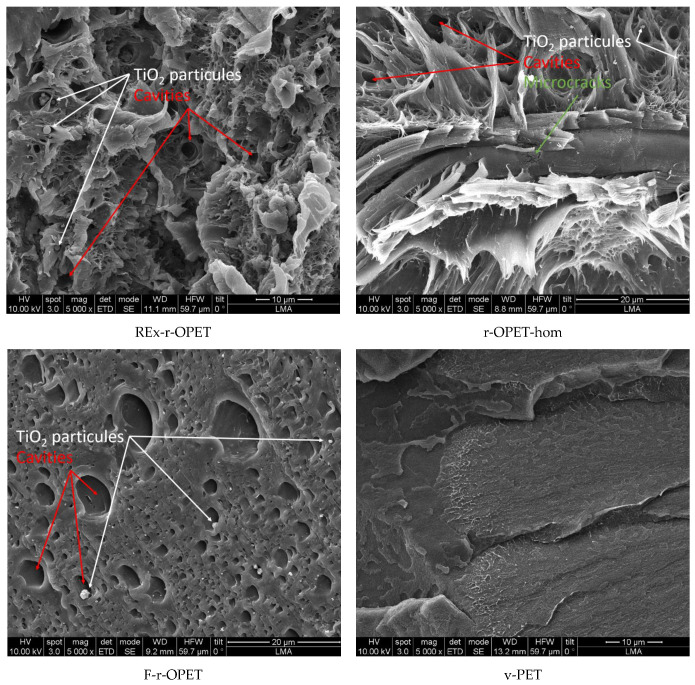
SEM images of fractured surfaces during T-T stressed at 95 %UTS (×5000).

**Figure 13 polymers-14-03466-f013:**
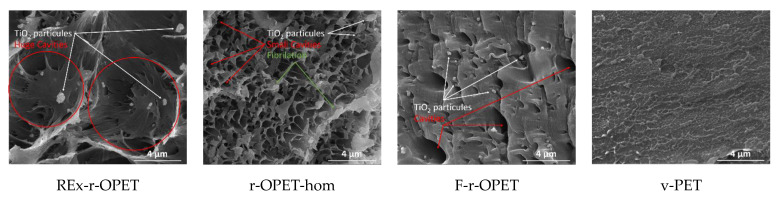
SEM images of fractured surfaces during T-T stressed at 95 %UTS (×20,000).

**Table 1 polymers-14-03466-t001:** Tensile properties (DIC) of the materials.

Sample	UTS (MPa)		Tensile Modulus (MPa)	
REx-r-OPET	49.6	±0.9	2286	±35
r-OPET-hom	47.9	±1.4	2169	±31
F-r-OPET	49.3	±0.5	2173	±160
v-PET	51.5	±0.5	2182	±67

**Table 2 polymers-14-03466-t002:** Relationships between applied stress amplitude and fatigue life by modelling.

Sample	Logarithmic Law	Basquin Law
REx-r-OPET	S = −8.1 log(N) + 189, R^2^ = 0.95	σ_α_ = 140.8(N_r_)^−0.096^, R^2^ = 0.94
r-OPET-hom	S = −7.1 log(N) + 169, R^2^ = 0.98	σ_α_ = 114.0(N_r_)^−0.087^, R^2^ = 0.97
F-r-OPET	S = −3.8 log(N) + 130, R^2^ = 0.94	σ_α_ = 70.0(N_r_)^−0.045^, R^2^ = 0.93
v-PET	S = −4.2 log(N) + 143, R^2^ = 0.94	σ_α_ = 83.2(N_r_)^−0.047^, R^2^ = 0.95

## Data Availability

The data presented in this study are available on request from the corresponding author.

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
