# Peer review of "Evaluation of Fatigue Life of Recycled Opaque PET from Household Milk Bottle Wastes"

_polymers, 2022, doi:10.3390/polym14173466_

Round 1
Reviewer 1 Report
The paper shows the results of a fatigue test investigation on post-consumer recycled opaque PET at various steps of the recycling process. Hypotheses on the failure mechanisms are advanced with the aid also of Scanning Electron Microscopy micrographs
In my opinion the title and the abstract (see lines 15-16 and 17-20) are misleading. One would expect the proposal of a new recycling process. On the contrary the new process is proposed in a previous paper. The present one is instead a characterization through fatigue test and SEM. I would suggest to modify the title and the abstract in order to better address the reader when downloading the paper
In the abstract (lines 20-21) the authors state that “correlations have been carried out…..of each type of PET” without having previously described which type they have investigated. Please specify better
The recycling process details should be described in the experimental section without obliging the reader to find them in another paper (see lines 182-187) that however is to be referenced so as the authors have already done.
Which are the post treatments the authors refer to on line 229?
The magnification should be indicated in the captions of Figg. 7 and 8 not only in the manuscript text
Author Response
Dear Editor, we are sincerely grateful to the reviewers for considering our work and for their useful comments on the original research article entitled “Enhancement of fatigue life of recycled opaque PET from household milk bottle wastes” by Adrian Korycki, Christian Garnier, Silvia Irusta and France Chabert, for consideration for publication in Polymers (Manuscript ID: polymers-1858142).
Our responses are presented below in blue. Hopefully, these answers will clarify the reviewer’s points. Some changes were done in the attached manuscript; they are highlighted in yellow.
The authors would like to thank the reviewer for his/her valuable comments that improved the quality of the manuscript. Below we address them one by one. The authors have done their best to improve the quality of the manuscript to meet the high standard of the journal.
Reviewer #1 (05/08/2022):
The paper shows the results of a fatigue test investigation on post-consumer recycled opaque PET at various steps of the recycling process. Hypotheses on the failure mechanisms are advanced with the aid also of Scanning Electron Microscopy micrographs.
Response: We would like to thank the reviewer for the thorough review.
In my opinion the title and the abstract (see lines 15-16 and 17-20) are misleading. One would expect the proposal of a new recycling process. On the contrary the new process is proposed in a previous paper. The present one is instead a characterization through fatigue test and SEM. I would suggest to modify the title and the abstract in order to better address the reader when downloading the paper.
Response: The word “enhancement” has been replaced by “evaluation” in the title to remove the idea of a new recycling process. Special attention was given to the abstract section by rewording some sentences as suggested by the reviewers to clarify some points. Lines 15-21
In the abstract (lines 20-21) the authors state that “correlations have been carried out…..of each type of PET” without having previously described which type they have investigated. Please specify better.
Response: The different types of specimens have been mentioned in the abstract to clarify this point. Lines 19-21
The recycling process details should be described in the experimental section without obliging the reader to find them in another paper (see lines 182-187) that however is to be referenced so as the authors have already done.
Response: Thank you for your comment. We have described the homogenization step by extrusion and the reactive extrusion process with a chain extender in the Materials section, to obtain r-OPET-hom and REx-r-OPET, respectively. Lines 194-215
Which are the post treatments the authors refer to on line 229?
Response: Thank you for your remark, the post-treatment considers the images obtained by digital image correlation. The sentence has been corrected: “Tensile test runs and post-treatment of DIC images were done for at least 3 samples”. Line 260
The magnification should be indicated in the captions of Figg. 7 and 8 not only in the manuscript text.
Response: Thank you for your comment, we have added a scale of the magnifications on the SEM images (now Fig. 10 and 11), which is more visible now.
Reviewer 2 Report
In this paper the writer demonstrated a good effort in upcycling opaque PET milk bottles and enhanced their fatigue life. This might be a beneficial step in minimizing the plastics waste from our environment, apart from saving resources.
However some corrections and suggestions are need to be followed.
1. The writer must include a graphical representation of the percent use of PET in different household items including milk bottles.
2. The literature survey looks incomplete without adding some visual adds, facts and figures about the recycling precesses, total amount in M.Tons being recycled and being wasted for the last 10 years.
3. Fig 4 and Fig 5. looks odd and cannot attract the readers. Further these figures reflects the core results of this paper so must be presented clearly and vividly instead of using dots and symbols.
4. In Fig 9 and 10, the writer must use arrows to guide the readers about the presence of TiO2 particals, bright spots, and spherical cavities preset and being explained in SEM images.
5. Please provide the EDX results, showing the amount of species like TiO2 and others present in PET and other samples.
Author Response
Dear Editor,
we are sincerely grateful to the reviewer for considering our work and for his/her useful comments on the original research article entitled “Enhancement of fatigue life of recycled opaque PET from household milk bottle wastes” by Adrian Korycki, Christian Garnier, Silvia Irusta and France Chabert, for consideration for publication in Polymers (Manuscript ID: polymers-1858142).
Our responses are presented below in blue. Hopefully, these answers will clarify the reviewer’s points. Some changes were done in the attached manuscript; they are highlighted in yellow.
The authors would like to thank the reviewer for his/her valuable comments that improved the quality of the manuscript. Below we address them one by one. The authors have done their best to improve the quality of the manuscript to meet the high standard of the journal.
Reviewer #2:
In this paper the writer demonstrated a good effort in upcycling opaque PET milk bottles and enhanced their fatigue life. This might be a beneficial step in minimizing the plastics waste from our environment, apart from saving resources.
Response: We would like to thank the reviewer for the thorough review.
However some corrections and suggestions are need to be followed.
Response: We have gone through the comments one by one and addressed them as detailed below. We also improved the paper to meet the standard expected by the reviewer.
- The writer must include a graphical representation of the percent use of PET in different household items including milk bottles.
Response: Thank you for your remark. We have added a graphical representation of the percent use of PET in Europe. A new graph in Figure 2 represents market shares of PET in Europe (bottles, trays, film, strapping and fibre), as well as, market shares of PET by bottle types and colours, milk bottles represent 2 %. Lines 59-65
- The literature survey looks incomplete without adding some visual adds, facts and figures about the recycling precesses, total amount in M.Tons being recycled and being wasted for the last 10 years.
Response: Thank you for your comment, we added a new graph, Figure 1, which represents the evolution of post-consumer plastic waste treatment in Europe from 2006 to 2020. The graphic shows the evaluation of landfill, energy recovery and recycling). Lines 34-41
- Fig 4 and Fig 5. looks odd and cannot attract the readers. Further these figures reflects the core results of this paper so must be presented clearly and vividly instead of using dots and symbols.
Response: We agree with the reviewer’s remark. However, we have chosen this representation in Fig 4 and 5, (now Figure 7 and 8) to show that fatigue tests need a large number of samples and it gives a high dispersion of the results. This representation is common in fatigue studies. We have added some explanations in the text as well as in the legend for a better understanding. Line 320-329
- In Fig 9 and 10, the writer must use arrows to guide the readers about the presence of TiO2 particals, bright spots, and spherical cavities preset and being explained in SEM images.
Response: Thank you for your suggestion. We have added some arrows and identifications to guide the readers in Figure 12. TiO2 particles, cavities, microcracks and fibrillation have been pointed out to clarify the images.
- Please provide the EDX results, showing the amount of species like TiO2 and others present in PET and other samples.
Response: Please see the attached file
